# Transmetatarsal amputations in patients with diabetes mellitus: A contemporary analysis from an academic tertiary referral centre in a developing community

Qusai Aljarrah[1]*, Mohammed Z. Allouh[2,3]*, Anas Husein[1], Hussam Al-Jarrah[1], Amer Hallak[4], Sohail Bakkar[5], Hamzeh Domaidat[1], Rahmeh Malkawi[5]

1 Department of General & Vascular Surgery, Faculty of Medicine, Jordan University of Science and Technology, Irbid, Jordan, 2 Department of Anatomy, College of Medicine and Health Sciences, United Arab Emirates University, Al Ain, United Arab Emirates, 3 Department of Anatomy, Faculty of Medicine, Jordan University of Science and Technology, Irbid, Jordan, 4 Faculty of Medicine, Jordan University of Science and Technology, Irbid, Jordan, 5 Department of Surgery, Faculty of Medicine, The Hashemite University, Zarqa, Jordan

* qmaljarrah@just.edu.jo (QA); m_allouh@uaeu.ac.ae (MZA)

## Abstract

Transmetatarsal amputation (TMA) involves the surgical removal of the distal portion of metatarsals in the foot. It aims to maintain weight-bearing and independent ambulation while eliminating the risk of spreading soft tissue infection or gangrene. This study aimed to explore the risk factors and surgical outcomes of TMA in patients with diabetes at an academic tertiary referral center in Jordan. Medical records of all patients with diabetes mellitus who underwent TMA at King Abdullah University Hospital, Jordan, between January 2017 and January 2019 were retrieved. Patient characteristics along with clinical and laboratory findings were analyzed retrospectively. Pearson's chi-square test of association, Student's *t*-test, and multivariate regression analysis were used to identify and assess the relationships between patient findings and TMA outcome. The study cohort comprised 81 patients with diabetes who underwent TMA. Of these, 41 (50.6%) patients achieved complete healing. Most of the patients were insulin-dependent (85.2%). Approximately half of the patients (45.7%) had severe ankle-brachial index (ABI). Thirty patients (37.1%) had previous revascularization attempts. The presence of peripheral arterial disease ($P<0.05$) exclusively predicted poor outcomes among the associated comorbidities. Indications for TMA included infection, ischemia, or both. The presence of severe ABI ($\leq 0.4$, $P<0.01$) and a previous revascularization attempt ($P<0.05$) were associated with unfavorable outcomes of TMA. Multivariate analysis that included all demographic, clinical, and laboratory variables in the model revealed that insulin-dependent diabetes, low albumin level (< 33 g/L), high C-reactive protein level (> 150 mg/L), and low score of Laboratory Risk Indicator for Necrotizing Fasciitis (LRINEC, <6) were the main factors associated with poor TMA outcomes. TMA is an effective technique for the management of diabetic foot infection or ischemic necrosis. However, attention should be paid to certain

**Data Availability Statement:** All relevant data are within the paper and its Supporting Information files.

**Funding:** The authors received no specific funding for this work.

**Competing interests:** The authors have declared that no competing interests exist.

important factors such as insulin dependence, serum albumin level, and LRINEC score, which may influence the patient's outcome.

## Introduction

Transmetatarsal amputation (TMA) is a surgical technique that involves the removal of a part of the foot, which includes the distal portion of the metatarsals [1]. It is a relatively common surgery performed to treat a severely infected or ischemic foot [2]. The aims of TMA are to maintain weight-bearing and independent ambulation while eliminating the risk of spreading soft tissue infection or gangrene. Published literature indicates that three out of four transmetatarsal amputations (TMAs) are performed for diabetic foot complications [3]. In a contemporary study among US veterans, TMA accounted for approximately one-third of the current increase in total lower extremity amputation (LEA) [4]. In developing nations with limited rehabilitation services, not only is TMA a limb salvage operation, but it also contributes to functional independence [5].

Since the first TMA was performed in the 19th century, it inherited an unfavorable reputation due to wound-related morbidity and inconsistent healing outcomes reported in published literature, particularly over the last two decades [1, 6–9]. The aging population with diabetes and atherosclerosis is an emerging global health care challenge [10]. In Jordan, the prevalence of diabetes is 23.7% which represents an 83% increase over the last two decades [11]. Diabetes is associated with a high rate of lower limb adverse effects and non-traumatic LEA, where diabetic foot-related complications remain one of the most common causes of hospitalization among various other systemic complications of diabetes [12, 13]. Published TMA healing rates are primarily extrapolated from western data with scarce data from developing countries. Hence, this analysis is the first to explore the outcomes of non-traumatic TMA in the Middle East and North Africa (MENA). Identification of predictors of healing in our data is essential for direct effective service delivery to curb poor outcomes associated with TMA.

Identifying patient variables associated with TMA outcomes is challenging and controversial in published literature [14–16]. Furthermore, patients have different perceptions of limb loss interventions; while some patients opt for definitive major LEA, and others require some time to gain the expected limb loss during the treatment journey of TMA 'Stairway to Amputation'. Current studies emphasize that clinical acumen remains the most important factor in patient selection for TMA due to uncertainty with existing objective perioperative measures of wound healing [17, 18]. A recent meta-analysis found that the reamputation rates surprisingly remained unchanged over the past two decades despite current cutting-edge therapies [19].

This study aimed to review the associated risk factors and outcomes of TMA in diabetic patients in Jordan as a developing nation. Additionally, it aimed to identify the main predictors of TMA failure in these patients. Identifying the predictors of failure of the TMA procedure is crucial to select the best definitive amputation for such a comorbid population, and to avoid unnecessary theater trips.

## Methods

### Study protocol

This retrospective study included all diabetic patients who underwent TMA at King Abdullah University Hospital (KAUH) in Northern Jordan from January 2017 to January 2019. KAUH is a tertiary referral center in northern Jordan with a capacity of 683 beds. This teaching

hospital is affiliated with the Jordan University of Science and Technology (JUST). The study was conducted with the approval of the Institutional Review Board Committee at JUST, and was conducted in accordance with the principles of the Declaration of Helsinki and its later amendments for ethical research performance. Patient consent was not required because the data were used in aggregates with no personal identifiers. Patients who underwent traumatic TMA were excluded from the study.

Data were abstracted from the hospital electronic records and included patient demographics (i.e., sex and age), comorbidities, laboratory values, as well as peri, - and postoperative clinical findings. All laboratory test results were obtained within 24 hours of hospital admission. Laboratory tests included blood serum levels of hemoglobin (Hb), albumin, C-reactive protein, and hemoglobin A1c. We also analyzed the effectiveness of the Laboratory Risk Indicator for Necrotizing Fasciitis (LRINEC) score in predicting healing potential in TMA for diabetic foot. The LRINEC score is a 13-point scoring system based on routine laboratory indicators including white blood cell count, along with C-reactive protein (CRP), hemoglobin, sodium, creatinine, and glucose levels [20].

Indications for TMA were forefoot sepsis, forefoot ischemia, or a combination of sepsis and ischemia. Forefoot sepsis was diagnosed clinically based on the presence of at least two classic findings of inflammation or purulence [6]. The clinical absence of palpable pedal pulses in the ipsilateral limb indicated forefoot ischemia and the presence of peripheral artery disease (PAD). Ankle-brachial pressure index (ABI) was recorded for all patients and was used to assess the degree of severity of PAD as either mild (0.7–0.9), moderate (0.41–0.69), or severe (≤0.4). TMA was considered healed when complete re-epithelialization of the surgical wound occurred. Chronic stump ulceration at 1 year or revision to major LEA are considered a failure of TMA.

The length of stay (LOS) was calculated from the time of admission to hospital discharge. All TMAs were performed by vascular surgeons at our institute. Surgical techniques and perioperative management were identical in all cases.

## Surgical technique

All TMA procedures were performed under regional anesthesia (ankle block). A standard fish mouth incision was made at the most distal aspect of the grossly healthy tissue to create the upper and lower flaps. Meticulous sharp dissection with removal of all devitalized, infected tissue, avascular structures, and division of all exposed tendons after pulling to the maximum length. Electrocautery was avoided in all our cases, and hemostasis was secured by direct pressure or suture ligation of the bleeding vessel. Bone edges were refreshed, and obtaining samples for tissue culture was essential in all infected cases. Extensive wound irrigation and washout with careful inspection of flap edges along with final trimming to an adequate length were done. All TMA wounds in this study were kept open to heal by secondary intention or delayed primary closure (staged closure when sepsis was controlled and sloughy tissue was eliminated). Antimicrobial therapy was initiated empirically according to institutional guidelines and adjusted according to the culture and sensitivity tests. The use of advanced wound products and negative pressure therapy was individualized according to case requirements [21]. Meticulous wound management was performed by a dedicated tissue viability team at our institute.

## Statistical analysis

The factors that were investigated in relation to TMA were described using frequency distribution for categorical variables and mean ± standard deviation for continuous variables.

Pearson's chi-square (χ2) test was used to analyze the associations between categorical variables, and Student's *t*-test was used for continuous variables. In addition, binary logistic regression analysis was used to determine the main predictors of TMA failure in the study model. Statistical significance was set at $P<0.05$. A *post-hoc* residual analysis was also conducted to determine the exact significance in the contingency tables.

## Results

The study cohort consisted of 81 patients with diabetes who underwent non-traumatic TMA. The patient characteristics and clinical presentations are summarized in Table 1. The mean age of the patients was 63.8 ± 13.5 years, and approximately three-quarters were male patients. About half of the patients (50.6%) were obese, and more than one-third (37.0%) were overweight. Most patients had insulin-dependent diabetes (85.2%). Approximately half of the patients (45.7%) had severe ABI scores. Thirty patients (37.1%) had a previous revascularization attempt with a mean time of 34 days for revascularization, and 29 (35.8%) had a previous toe amputation (Table 1). About half of the patients with TMA had completely healed (50.6%), while others either progressed to a proximal amputation (24.7%), had a chronic stump ulcer (13.6%), died within 30 days of amputation (6.2%), or were lost to follow-up (4.9%). The mean length of hospital stay was 19.2 ± 13.2 days.

### Factors associated with failed TMA

The independent factors that might be associated with the outcome of TMA in diabetic patients are summarized in Table 2. The presence of PAD ($P<0.05$), ABI ($P<0.01$), and the incidence of a previous revascularization attempt ($P<0.05$) were linked to a significant failure in TMA. Patients with a normal ABI were more likely to heal ($P<0.05$), while patients with a severe ABI were more likely to fail TMA ($P<0.01$). Surprisingly, patients who underwent previous ipsilateral revascularization procedures were more likely to fail than those who did not ($P<0.05$).

In the multivariate logistic regression analysis model that included all factors in Table 2, the main predictors that were associated with failed TMA procedure included insulin dependence, low albumin levels, high CRP levels, and low LRINEC score (Table 3). Patients who were dependent on insulin had a significantly (482.5 times) higher risk of TMA failure than non-insulin-dependent patients ($P<0.05$). Additionally, patients with albumin levels of below 33 g/L had a significantly (~283 times) higher risk of TMA failure than patients with normal albumin levels of between 34 and 54 g/L ($P<0.01$). Patients with CRP levels of above 150 mg/L had a significantly (~162 times) higher risk of TMA failure than patients with CRP levels of below 150 mg/L ($P<0.05$). Interestingly, patients with LRINEC scores of <6 had a 749.5 times greater risk of TMA failure than patients with LRINEC score of ≥ 6 ($P<0.01$).

### Factors associated with revision to major LEA

The analysis of the independent factors that might be associated with TMA revision to major LEA was almost comparable to the factors associated with TMA failure (Table 4). The presence of PAD ($P<0.05$), ischemia ($P<0.05$), severe ABI ($P<0.01$), and a previous revascularization attempt ($P<0.05$) were all associated with the progression of TMA to major LEA. Obese patients tended to progress to major LEA; however, this finding was at the limit of significance ($P = 0.055$, adjusted residuals).

In the multivariate logistic regression analysis model that included the factors in Table 4, the main predictors associated with progression to major LEA were insulin dependence, low albumin levels, and low LRINEC score (Table 5). Patients who were insulin dependent had a

**Table 1. Characteristics and clinical presentations of diabetic patients who underwent transmetatarsal amputation (TMA).**

| Associated variables | Number | Percent (%) |
|---|---|---|
| | | Mean ± SD |
| **Total Patients** | 81 | 100.0 |
| **Sex** | | |
| Male | 60 | 74.1 |
| Female | 21 | 25.9 |
| **Age (y)** | 63.8 ± 13.5 | |
| **BMI** | | |
| Healthy (18.5–24.9) | 10 | 12.3 |
| Overweight (25.0–29.9) | 30 | 37.0 |
| Obese (≥ 30.0) | 41 | 50.6 |
| **Comorbidities** | | |
| Hypertension | 56 | 69.1 |
| IHD | 31 | 38.3 |
| Hypercholesterolemia (>5.2 mmol/L) | 6 | 7.4 |
| ESRD | 9 | 11.1 |
| Peripheral Artery Disease | 50 | 61.7 |
| **Insulin Dependent Patients** | 69 | 85.2 |
| **Smoking** | 49 | 60.5 |
| **Laboratory findings** | | |
| Low Hb (<11.0 g/dL) | 51 | 63.0 |
| Low albumin (<33 g/L) | 38 | 46.9 |
| High HbA1c (>8.0%) | 56 | 69.1 |
| High CRP (>150 mg/L) | 36 | 44.4 |
| **Indication for TMA** | | |
| Infection | 31 | 38.3 |
| Ischemia | 33 | 40.7 |
| Combined | 17 | 21.0 |
| **ABI** | | |
| Normal (0.91–1.30) | 12 | 14.8 |
| Mild (0.70–0.90) | 13 | 16.0 |
| Moderate (0.41–0.69) | 14 | 17.3 |
| Severe (≤ 0.40) | 37 | 45.7 |
| Missing | 5 | 6.2 |
| **Patients with previous revascularization** | | |
| Angioplasty | 22 | 27.2 |
| Bypass | 8 | 9.9 |
| **Patients with previous toe amputation** | 29 | 35.8 |
| **Surgery Outcome** | | |
| Healed | 41 | 50.6 |
| Stump ulcer | 11 | 13.6 |
| Proximal amputation | 20 | 24.7 |
| Dead | 5 | 6.2 |
| Unknown | 4 | 4.9 |
| **Length of stay (days)** | 19.2 ± 13.2 | |

(*Continued*)

**Table 1.** (Continued)

| Associated variables | Number | Percent (%) |
|---|---|---|
| | Mean ± SD | |
| LRINEC score on admission ($\geq$ 6) | 49 | 60.5 |

Abbreviations: ABI, ankle-brachial pressure index; BMI, body mass index; CRP, c-reactive protein; ESRD, end-stage renal disease; Hb, hemoglobin; IHD, ischemic heart disease; LRINEC, laboratory risk indicator for necrotizing fasciitis; SD, standard deviation; y, years.

significantly ($1.5 \times 10^4$ times) higher risk of progression to major LEA than that of non-insulin-dependent patients ($P<0.05$). Additionally, patients with albumin levels of below 33 g/L had a significantly (~301 times) higher risk of progression than patients with normal albumin levels of between 34 and 54 g/L ($P<0.05$). Lastly, patients with LRINEC scores <6 had a 623.6 times greater risk of progressing to major LEA than patients with LRINEC score of $\geq$ 6 ($P<0.05$).

## Discussion

To the best of our knowledge, this is the first study in Jordan to analyze the outcome predictors of TMA in patients with diabetes. The presence of PAD, indication for TMA, and ABI value were the main independent predictors of TMA outcome. However, in a multiple regression model that included all factors together, it was revealed that the main indicators that could predict the outcome of TMA were insulin dependence, laboratory values of albumin and CRP, and the LRINEC score.

Published failure rates of TMA range from 14% to 78% [16, 22–41]. Multiple confounders might explain discordant healing rates in published data: First, heterogeneity in studied patient cohorts, including diabetic and nondiabetic patients, or cases of traumatic TMA. Second, the sample size along with the time frame of analysis. For example, a United Kingdom-based retrospective review over a 12-year (more than a decade ago) reported only 54 TMA (4.5 amputations/ year) with a 78% healing rate [24]. Third, many patients have a pessimistic perception of major LEA, especially when physical independence is lost, and ambulation potential is minimized (functional salvage vs. limb salvage) due to primitive rehabilitation services. Therefore, TMA is sometimes performed as a stairway to the major LEA to allow for time to grieve a potential limb loss. Lastly, TMA is performed under local anesthesia; hence, it is sometimes offered to clinically debilitated patients when physiological reserve prohibits definitive interventions, such as revascularization or major LEA. Furthermore, it is worthy of note that most cases are admitted via the emergency department and surgeries are performed by residents during on-call hours when consultant supervision is not optimal.

The ipsilateral revision rate of TMA to major LEAs in our analysis was 27.8% (20/72). This is in accordance with revision rates in a recent systematic review by Thorud et al. [25], which indicated that 1/3 of patients who undergo TMA progressed to ipsilateral major LEA. In the current analysis, PAD with a severe ABI predicted proximal amputation since 85% of our patients who underwent major LEA had an ABI of $\leq$0.4 ($P = 0.003$), with ischemia being an indication for TMA in 60% of patients who progressed to major LEA ($P = 0.024$). Marston et al. [42] reported consistent results with our findings and mentioned that ABI was independently associated with amputation at 1 year, with 32% and 43% of limbs with an ABI of <0.5 and <0.4, required amputation respectively. Zhang et al. [38] reported that healing was achieved in 1/3 of moderately ischemic patients. They established that patients with a higher

**Table 2. Factors associated with the outcome of transmetatarsal amputation (TMA) in diabetic patients.**

| Outcome | Healed | Unhealed | P-value |
|---|---|---|---|
| | N (% from healed) | N (% from failed) | |
| **Compared Patient Groups** | 41 (100.0) | 31 (100.0) | – |
| **Sex** | | | |
| Male | 29 (70.7) | 24 (77.4) | NS |
| Female | 12 (29.3) | 7 (22.6) | |
| **Age (y),** mean ± SD | 62.7 ± 12.7 | 63.5 ± 15.5 | NS |
| **BMI,** mean ± SD | 29.5 ± 4.3 | 30.9 ± 5.8 | NS |
| **BMI** | | | |
| Healthy (18.5–24.9) | 6 (14.6) | 4 (12.9) | NS |
| Overweight (25.0–29.9) | 17 (41.5) | 8 (25.8) | |
| Obese (≥ 30.0) | 18 (43.9) | 19 (61.3) | |
| **Comorbidities** | | | |
| Hypertension | 25 (61.0) | 23 (74.2) | NS |
| IHD | 14 (34.1) | 13 (41.9) | NS |
| High Cholesterol (>5.2 mmol/L) | 2 (4.9) | 3 (9.7) | NS |
| ESRD | 3 (7.3) | 3 (9.7) | NS |
| Peripheral Artery Disease | 19 (46.3) | 22 (71.0)[↑] | 0.037[*] |
| **Insulin dependent patients** | 34 (82.9) | 27 (87.1) | NS |
| **Smoking** | 23 (56.1) | 21 (67.7) | NS |
| **Laboratory findings** | | | |
| Low Hb (<11.0 g/dL) | 23 (61.0) | 21 (67.7) | NS |
| Low albumin (<33 g/L) | 15 (36.6) | 18 (58.1) | NS |
| High HbA1c (>8.0%) | 32 (78.0) | 21 (67.7) | NS |
| High CRP (>150 mg/L) | 21 (51.2) | 14 (45.2) | NS |
| **Indication for TMA** | | | |
| Infection | 22 (53.7)[↑] | 9 (29.0) | 0.037[#] |
| Ischemia | 11 (26.8) | 15 (48.4) | |
| Combined | 8 (19.5) | 7 (22.6) | |
| **ABI** | | | |
| Normal (0.91–1.30) | 10 (24.4)[↑] | 2 (6.5) | 0.006[#] |
| Mild (0.70–0.90) | 7 (17.1) | 3 (9.7) | |
| Moderate (0.41–0.69) | 8 (19.5) | 4 (12.9) | |
| Severe (≤ 0.40) | 13 (31.7) | 20 (64.5)[↑↑] | |
| Missing | 3 (7.3) | 2 (6.5) | |
| **Previous revascularization** | 11 (26.8) | 16 (51.6)[↑] | 0.031[#] |
| **Previous toe amputation** | 11 (26.8) | 13 (41.9) | NS |
| **Length of stay (d),** mean ± SD | 20.8 ± 13.9 | 19.9 ± 12.9 | NS |
| **LRINEC score on admission (≥ 6)** | 30 (73.2) | 17 (54.8) | NS |

Abbreviations: ABI, ankle brachial pressure index; BMI, body mass index; CRP, c-reactive protein; ESRD, end-stage renal disease; Hb, hemoglobin; IHD, ischemic heart disease; LRINEC, laboratory risk indicator for necrotizing fasciitis; N, number; NS, not significant; P, probability; SD, standard deviation; y, years.

[↑]($P<0.05$)

[↑↑]($P<0.01$): significantly higher than expected frequency.

[*]P-value calculated using Pearson's chi-square test.

[#]P-value calculated using the adjusted residual analysis.

**Table 3. Multivariant analysis for factors associated with failed transmetatarsal amputations in diabetic patients.**

| | P-value | OR | 95% CI |
|---|---|---|---|
| **Insulin** | | | |
| Yes | 0.023 | 482.5 | $2.4–9.8 \times 10^4$ |
| No | Reference | | |
| **Albumin** | | | |
| Low (< 33 g/L) | 0.004 | 282.9 | $5.9–1.4 \times 10^4$ |
| Normal (34–54 g/L) | Reference | | |
| **C-Reactive Protein** | | | |
| > 150 mg/L | 0.015 | 162.5 | $2.7–9.8 \times 10^3$ |
| ≤ 150 mg/L | Reference | | |
| **LRINEC Score on Admission** | | | |
| < 6 | 0.009 | 749.5 | $5.4–1.0 \times 10^5$ |
| ≥ 6 | Reference | | |

Abbreviations: P, probability; OR, odds ratio; CI, confidence interval. The multivariate logistic regression analysis included the variables reported in Table 2.

ABI exhibited a higher probability of wound healing. Correspondingly, Pinzur et al. [43] reported a 92.2% healing rate after TMA with a minimum ABI of 0.5 in diabetic patients with serum albumin of 30 g/L. Hosch et al. [33] found that the most predictive factor for failed TMA was the established indication for surgery and only infected individuals with no underlying PAD were significantly more likely to heal at the level of the foot. However, Younger et al. [37] reported a poor correlation between vascularity and outcome. Also, Anthony et al. [40] conducted a comprehensive investigation and found no correlation between ABI and proximal LEA in individuals with TMA.

Despite being the recommended initial non-invasive test to detect PAD, the utilization of ABI as a screening tool for PAD is particularly inconsistent among diabetic patients when 58–84% of diabetic patients with significant PAD have elevated ABI values [44]. In addition, an extensive review article elucidated a fluctuating performance of ABI <0.9 (sensitivity, 29–95%, median at 63%; and specificity, 58–97%, median 93%) [45]. The unreliability of ABI in patients with diabetes may be attributed to the existence of peripheral diabetic neuropathy, medial arterial calcification, and/or incompressible arteries. Moreover, ABI is operator-dependent, rendering it inaccurate in some cases [46]. In diabetic patients, the toe brachial index (TBI) may overcome unreliable ABI and predict healing potential [47]. Unfortunately, our service does not have a TBI facility and the published data remains conflicting regarding the utility of TBI in predicting TMA outcomes.

The findings of this study are comparable to previous studies that palpable pedal (pedal) pulses are positive predictors of TMA healing [16, 41]. In fact, our data revealed that only one patient with a palpable pedal pulse progressed to major LEA. Moreover, not only did severe ABI predict proximal amputation in our analysis, but also predicted healing potential at 1 year following TMA ($P = 0.006$). The dilemma of attempting to predict reamputation risk among diabetic patients embraces the lack of individualized risk assessment, as developed models tend to inform population risk in general. In a recent study, a novel reamputation risk prediction model (AMPREDICT Reamputation) was thoroughly validated as a possible solution [48]. The AMPREDICT model can be used to quantify the individual risk of reamputation within 1 year of amputation indicated by diabetes and/or PAD complications and evaluate those who survive the first year after the incident amputation [48].

**Table 4. Factors associated with the transmetatarsal amputation (TMA) patients who progressed to proximal amputation.**

| Outcome | Healed | Progressed to proximal amputation | P-value |
|---|---|---|---|
| | N (% from healed) | N (% from failed) | |
| **Compared Patient Groups** | 41 (100.0) | 20 (100.0) | – |
| **Sex** | | | |
| Male | 29 (70.7) | 17 (85.0) | NS |
| Female | 12 (29.3) | 3 (15.0) | |
| **Age (y),** mean ± SD | 62.7 ± 12.7 | 61.2 ± 13.4 | NS |
| **BMI,** mean ± SD | 29.4 ± 4.3 | 32.0 ± 5.4 | NS |
| **BMI** | | | |
| Healthy (18.5–24.9) | 6 (14.6) | 2 (10.0) | 0.055[#] |
| Overweight (25.0–29.9) | 17 (41.5) | 4 (20.0) | |
| Obese (≥ 30.0) | 18 (43.9) | 14 (70.0) | |
| **Comorbidities** | | | |
| Hypertension | 25 (61.0) | 14 (70.0) | NS |
| IHD | 14 (34.1) | 7 (35.0) | NS |
| High Cholesterol (>5.2 mmol/L) | 2 (4.9) | 2 (10.0) | NS |
| ESRD | 3 (7.3) | 3 (15.0) | NS |
| Peripheral Artery Disease | 19 (46.3) | 16 (80.0)[↑] | 0.013* |
| **Insulin dependent patients** | 34 (82.9) | 18 (90.0) | NS |
| **Smoking** | 23 (56.1) | 16 (80.0) | NS |
| **Laboratory findings** | | | |
| Low Hb (<11.0 g/dL) | 25 (61.0) | 12 (60.0) | NS |
| Low albumin (<33 g/L) | 15 (36.6) | 10 (50.0) | NS |
| High HbA1c (>8.0%) | 32 (78.0) | 14 (70.0) | NS |
| High CRP (>150 mg/L) | 21 (51.2) | 9 (45.0) | NS |
| **Indication for TMA** | | | |
| Infection | 22 (53.7)[↑] | 4 (20.0) | 0.024* |
| Ischemia | 11 (26.8) | 12 (60.0)[↑] | |
| Combined | 8 (19.5) | 4 (20.0) | |
| **ABI** | | | |
| Normal (0.91–1.30) | 10 (24.4) | 1 (5.0) | 0.003* |
| Mild (0.70–0.90) | 7 (17.1) | 0 (0.0)[↓] | |
| Moderate (0.41–0.69) | 8 (19.5) | 2 (10.0) | |
| Severe (≤ 0.40) | 13 (31.7) | 17 (85.0)[↑↑] | |
| Missing | 3 (7.3) | 0 (0.0) | |
| **Previous revascularization** | 11 (26.8) | 12 (60.0)[↑] | 0.012* |
| **Previous toe amputation** | 11 (26.8) | 6 (30.0) | NS |
| **Length of stay (d),** mean ± SD | 20.8 ± 13.9 | 22.9 ± 13.6 | NS |
| **LRINEC score on admission (≥ 6)** | 30 (73.2) | 11 (55.0) | NS |

Abbreviations: ABI, ankle brachial pressure index; BMI, body mass index; CRP, c-reactive protein; ESRD, end-stage renal disease; Hb, hemoglobin; IHD, ischemic heart disease; LRINEC, laboratory risk indicator for necrotizing fasciitis; N, number; NS, not significant; P, probability; SD, standard deviation; y, years.

↑($P<0.05$)

↑↑($P<0.01$): significantly higher than expected frequency.

*$P$-value calculated using Pearson's chi-square test.

#$P$-value calculated using the adjusted residual analysis.

**Table 5. Multivariant analysis for factors associated with progression to major LEA.**

| | *P*-value | OR | 95% CI |
|---|---|---|---|
| **Insulin** | | | |
| Yes | 0.020 | $1.5 \times 10^4$ | $4.7–5.0 \times 10^7$ |
| No | | Reference | |
| **Albumin** | | | |
| Low (< 33 g/L) | 0.033 | 300.7 | $1.6–5.8 \times 10^4$ |
| Normal (34–54 g/L) | | Reference | |
| **LRINEC Score on Admission** | | | |
| < 6 | 0.047 | 623.6 | $1.1–3.5 \times 10^5$ |
| ≥ 6 | | Reference | |

Abbreviations: LEA, lower extremity amputation; P, probability; OR, odds ratio; CI, confidence interval. The multivariate logistic regression analysis included all variables in Table 4 except comorbidities and length of stay (LOS).

There is skepticism in the existing literature on the benefit of revascularization in patients undergoing TMA. Some authors identified positive results with concomitant revascularization [22], while others denied recording any benefit [38]. Toursarkissian et al. [47] indicated that failed TMA is a multifactorial process, and failed revascularization is not a major event that predicts TMA failure. However, the authors included a selected group in which revascularization was performed in 79.5% of TMA patients with a mean toe pressure of 31 mmHg. Furthermore, the same authors indicated that the healing potential could not be predicted using angiographic findings. In our analysis, previous revascularization imposed a negative outcome on healing; revascularization was associated with poor healing at 1 year ($P = 0.031$) and revision to proximal amputation ($P = 0.012$) in patients that underwent TMA. This paradox is multifactorial and may be explained by the following facts regarding diabetic vasculopathy. First, it is likely that patients who underwent vascular procedures had a greater burden of arterial occlusive disease (ABI≤0.4), which would predispose to poor healing after TMA. In a study from China [38], the authors explored the outcomes of TMA in diabetic patients who were not candidates for revascularization and concluded that satisfactory results can be achieved in patients with limb ischemia undergoing TMA; however, only 11.1% of their study cohort had severe ischemia with ABI of ≤0.4. Second, the lack of a timely approach to revascularization when the mean time to revascularization was 34 days in the current data. Moreover, futile attempts to achieve inline flow in cases of severe occlusive disease of foot vessels and pedal arch (Desert foot) renders tibial interventions suboptimal to secure adequate tissue healing and is infrequently linked to physical deconditioning. Finally, vascular and endovascular interventions in patients with diabetes are technically demanding, with a significant early failure rate. As reported by Mueller et al. [36], vascular reconstruction before TMA is not always protective, as 57% of patients who required a more proximal amputation had prior vascular reconstructive surgery. Therefore, the ideal reconstruction of complex foot defects remains suboptimal. Santanelli di Pompeo *et al*. identified critical anatomical components that incorporate microvascular reconstruction to achieve favorable outcomes in demanding wounds [49]. They identified a structural classification including a bony platform, soft tissue envelope, and defect size as fundamental components when approaching demanding foot defects. Also, previous studies demonstrated a substantial improvement in the microcirculation of DFU via decompression of the plantar neurovascular bundle and tarsal tunnel release [50–52]. Furthermore, they addressed the importance of diabetic neuropathy as a cornerstone factor in DFU outcomes [52].

Multivariate analysis revealed that insulin dependence, hypoalbuminemia (<33 g/L), elevated CRP (>150 mg/L), and LRINEC score of <6 were found to be associated with failed TMA. Our data revealed that about 85% of the studied cohort had insulin dependence and diabetes was uncontrolled (HbA1c >8.0%) in more than two-thirds of the patients. Interestingly, the Bypass Angioplasty Revascularization Investigation in type 2 diabetes (BARI-2D) trial demonstrated a lower incidence of PAD in non-insulin than in insulin dependent patients [53]. Furthermore, poor glycemic control is linked to an increased incidence of PAD and peripheral neuropathy, where a 1% increase in HbA1c was associated with a 28% increased incidence in PAD, while a 1% reduction in HbA1c was accompanied by a 43% reduction in the risk of amputation [54]. Moreover, peripheral neuropathy is often overlooked in our database, and regrettably, the authors did not explore the impact of diabetic neuropathy on TMA healing and re-ulceration.

Available data suggests that hypoalbuminemia is linked to adverse outcomes following TMA, which is tenable [38]. Zhang et al. [38] found that patients with a serum albumin level of ≥30 g/l had a 70% healing rate, whereas Hosch et al. [33] identified malnutrition as a predictor of poor healing following TMA. Elevated CRP levels are expected in any inflammatory or infectious processes, but extreme values of >150 mg/dl usually indicate a severe systemic inflammatory response and impaired glucose tolerance [55]. A retrospective analysis by Choi et al. [55] revealed that elevated CRP level was a predictor of failure of limb salvage interventions in patients with diabetic foot ulcer (DFU). Additionally, Pickwell et al. [56] compared various prediction and scoring models for DFU and identified CRP among other pertinent values as independent predictors of amputations in DFU.

In the current analysis, the LRINEC score was utilized as a numerical model to predict any trends in TMA outcomes as it combines multiple parameters related to wound healing. The counterintuitive inverse correlation between LRINEC score and healing potential can be easily explained by the fact that this score was principally developed to detect necrotizing fasciitis in soft tissue infections, and it lacks any parameter linked to vascular insufficiency, which is an important determinant of healing potential in DFU. In a recent study from Turkey [57], the authors explored the predictability of amputation or death in DFU according to the LRINEC score and showed that a score of ≥5 predicted amputations, while a score of ≥7 predicted mortality. In 2014, the WIFI system was developed by the Society for Vascular Surgery as a prognostic tool for predicting outcomes in patients with threatened diabetic limbs [58]. Interestingly, the WIFI system failed to predict the major LEA potential following TMA in a comparative study by Elsherif et al. [34]. Therefore, outcome predictive tools that incorporate parsimonious clinical, microbiological, and laboratory parameters, which are validated by data from vascular registries, should be developed to assist in making future decisions.

Hospital length of stay (LOS) remains an important determinant of healthcare tariffs. A recent analysis of the perioperative cost of major LEA in Jordan identified that LOS remains the principal contributor to the final toll of major LEA [59]. In the United States, the cost of diabetic foot-related complications exceeds healthcare expenditure on the five most common cancers in the United States. Corresponding data from the United Kingdom indicated that the NHS spent more on the management of diabetic foot than on breast, prostate, and lung cancers combined [60]. The mean LOS in our data (19.2±13.2) is approximately three times that of major LEA (6.8 ± 0.4) from published data in Jordan [61]. We attribute this to meticulous wound care and frequent debridement sessions required for TMA, particularly when none of our patients had primary wound closure. In addition, the lack of community-based tissue viability services in Jordan results in prolonged institutionalization for wound care purposes. Therefore, subsequent curtailing of complications would tremendously impact the healthcare system in terms of costs and effectiveness of treatment.

We acknowledge that our study has the inherent limitations of a typical retrospective analysis. It was also a single-centered study, with a moderate number of patients. However, we could not identify similar studies in the MENA region. We also did not explore the impact of timing or type of revascularization on the outcome of TMA. Furthermore, our study focused on perioperative parameters that can predict healing without focusing on other confounders, such as advanced wound care and ambulatory status. In addition, we did not study TMA admission or wound revision episodes. In Jordan, like other developing nations, the care of diabetic ulcers is fragmented, and unfortunately the concept of treating a "hole" in a patient rather than treating the "whole" patient is prevailing among care providers in our region. Consequently, in Jordan, patients present in a delayed fashion with an advanced ischemia or infection compared to cases seen in Western populations. Following an allied robust multidisciplinary approach can positively influence limb preservation. For example, in the United Kingdom, foot care provision services are integrated within the healthcare system. This has resulted in a significant inverse correlation with major diabetes-related lower limb amputation [62].

## Conclusions

In conclusion, some surgeons dispute the benefits of TMA trials before major LEA, especially in developing countries, since they do not have the indulgence of expensive interventions and little room for oversight [40]. However, we conclude that future studies on the burden of recurrent surgeries on patient deconditioning are essential. Patients with severe PAD are at an increased risk of adverse outcomes following TMA. Timely revascularization for patients with an ABI of ≤0.4 is of utmost interest when feasible, to allow effective positive outcome and avoid major future complications. In patients with an increased risk of failure, such as those with severe ischemia and unreconstructible vessels, definitive major LEA should be prudently considered to avoid inevitable failures. Conquering adverse outcomes by establishing predictive tools that explicate parsimonious clinical, microbiological, and laboratory parameters should be developed. These tools could help in predicting postoperative outcomes and assist in future decision making by the surgeon.

## Supporting information

**S1 Dataset.**
(XLSX)

## Author Contributions

**Conceptualization:** Qusai Aljarrah, Mohammed Z. Allouh.

**Data curation:** Anas Husein, Hussam Al-Jarrah.

**Formal analysis:** Mohammed Z. Allouh, Anas Husein, Hussam Al-Jarrah, Amer Hallak.

**Investigation:** Qusai Aljarrah, Mohammed Z. Allouh, Hamzeh Domaidat, Rahmeh Malkawi.

**Methodology:** Mohammed Z. Allouh, Hamzeh Domaidat.

**Project administration:** Qusai Aljarrah, Sohail Bakkar.

**Supervision:** Qusai Aljarrah, Mohammed Z. Allouh, Sohail Bakkar.

**Writing – original draft:** Qusai Aljarrah, Mohammed Z. Allouh, Amer Hallak, Rahmeh Malkawi.

**Writing – review & editing:** Mohammed Z. Allouh.

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
