## [Decision Letter · Decision Letter 0]

24 Aug 2022

PONE-D-21-26758Transmetatarsal Amputations in Patients with Diabetes Mellitus: A Contemporary Analysis from an Academic Tertiary Referral Centre in a Developing CommunityPLOS ONE

Dear Dr. ALJARRAH,

Thank you for submitting your manuscript to PLOS ONE. After careful consideration, we feel that it has merit but does not fully meet PLOS ONE’s publication criteria as it currently stands. Therefore, we invite you to submit a revised version of the manuscript that addresses the points raised during the review process.

ACADEMIC EDITOR: Please address questions raised by the reviewers, especially related to PAD and neuropathy. 

We look forward to receiving your revised manuscript.

Kind regards,

Tze-Woei Tan, M.D.

Academic Editor

PLOS ONE

Journal Requirements:

Additional Editor Comments (if provided):

Please address the comments and questions raised by the reviewers, especially related to PAD and neuropathy.

Reviewers' comments:

Reviewer's Responses to Questions

**Comments to the Author**

1. Is the manuscript technically sound, and do the data support the conclusions?

Reviewer #1: Yes

Reviewer #2: Yes

2. Has the statistical analysis been performed appropriately and rigorously? 

Reviewer #1: Yes

Reviewer #2: Yes

3. Have the authors made all data underlying the findings in their manuscript fully available?

Reviewer #1: No

Reviewer #2: Yes

4. Is the manuscript presented in an intelligible fashion and written in standard English?

Reviewer #1: Yes

Reviewer #2: Yes

5. Review Comments to the Author

Reviewer #1: the authors have done a fine job with the vascular component of this problem, but even though they mention diabetic neuropathy, they fail to discus its role in their failure. This should be done

Furthermore, is been shown that tarsal tunnel decompression of the four medial ankle tunes can improve sensation, prevent ulcer and amputation. The authors should discus the possibility of doing proximal nerve decompression at the same time as the amputaiton, and include these references. The decompression of the medial and lateral plantar vessels at the same time as the nerves may also improve blood flow.

Dellon, AL, The Four Medial Ankle Tunnels: A critical review of perceptions of tarsal tunnel syndrome and neuropathy, Neurosurg Clinics N America, 19:629-648, 2008.

Aszmann OC, Tassler PL, Dellon AL: Changing the natural history of diabetic neuropathy: Incidence of ulcer/amputation in the contralateral limb of patients with a unilateral nerve decompression procedure, Ann Plast Surg, 53:517-522, 2004.

Reviewer #2: Thank you for your submission. Congrats on your study. A few comments to further strengthen the paper:

- PAD is a significant aetiology for development of ischaemia hence TMA. There is a significant proportion of patients with low ABPI (mod/severe) in your cohort but the revascularisation rates are low. Would you be able to discuss more and comment? In addition, it will be interesting to see the effect of revascularisation efforts amongst the sub-group of patients with low ABPI. Counter intuitively, your results showed that patients with TMA failure and major LEA had higher percentages of revascularisation rather than those who healed. It's likely these patients had more severe PAD, rather than the act of revascularisation being a contributory factor

- LRINEC score is often used for necrotising fasciitis. Would you be able to elaborate why it was adopted in your population?

- What are your neuropathy rates?

- What is the time from TMA to full wound closure?

- Any wound adjuncts utilised? (e.g. NPWT or any advanced wound care)

- What is your off-loading / footwear regime? Stump ulcer may be due to poor footwear, rather than an actual TMA failure.

6. PLOS authors have the option to publish the peer review history of their article (what does this mean?). If published, this will include your full peer review and any attached files.

Reviewer #1: No

Reviewer #2: No

---

## [Author Response · Author response to Decision Letter 0]

16 Sep 2022

Response letter

Manuscript ID: PONE-D-21-26758

Manuscript Title: Transmetatarsal Amputations in Patients with Diabetes Mellitus: A Contemporary Analysis from an Academic Tertiary Referral Centre in a Developing Community

Dear Editor and Reviewers,

 We thank you for the comments provided to improve this manuscript. Changes have been made in the manuscript according to the comments. All corrections in the revised manuscript are highlighted in Yellow with blue text. Additionally, a detailed response to the comments is provided herewith. Please note that your comments are in black text, and our responses are in blue text.

Editor comments

The paper is very interesting and well written and gives some evidence on the risk factors connected to complicated TMA and progression to major LEA. Tables are very explicative. The discussion would benefit from adding a few sentences regarding improvement of microcirculation after tarsal tunnel release in diabetic patients, and possible forefoot microvascular reconstruction in motivated patients. The bibliography can be further implemented. The paper can be accepted after minor revision.

Page 4, line 9

Please add sequent citation form the same Journal

Coralie Amadou 1 2, Pierre Denis 3, Kristel Cosker 3, Anne Fagot-Campagna 3 Less amputations for diabetic foot ulcer from 2008 to 2014, hospital management improved but substantial progress is still possible: A French nationwide study- PLoS One. 2020 Nov 30;15(11):e0242524.

doi: 10.1371/journal.pone.0242524. eCollection 2020. PMID: 33253241, PMCID: PMC7703996, DOI: 10.1371/journal.pone.0242524

Response:

The reference has been added. The new reference is now reference # 13. Please refer to page 4, line 94 of the text for the corresponding citation.

Page 6, line 13

Please add following citation

Low-vacuum negative pressure wound therapy protocol for complex wounds with exposed vessels. Paolini G, Sorotos M, Firmani G, Gravili G, Ceci D, Santanelli di Pompeo F. J Wound Care. 2022 Jan 2;31(1):78-85. doi: 10.12968/jowc.2022.31.1.78.PMID: 35077217

Response:

The reference has been added. The new reference is now reference # 21. Please refer to page 7, line 162 of the text for the corresponding citation.

Page 12 line 14, 

Please Add citation and briefly discuss report about improvement of microcirculation after tarsal tunnel release in diabetic patients.

Evaluation of peripheral microcirculation improvement of foot after tarsal tunnel release in diabetic patients by transcutaneous oximetry.

Trignano E, Fallico N, Chen HC, Faenza M, Bolognini A, Armenti A, Santanelli Di Pompeo F, Rubino C, Campus GV.Microsurgery. 2016 Jan;36(1):37-41. doi: 10.1002/micr.22378. Epub 2015 Jan 13.PMID: 25641727

Response:

Please refer to our response to the comment below.

Page 12 line 14, 

Please add citation and briefly discuss report about possible forefoot microvascular reconstruction also in diabetic patients

Microvascular reconstruction of complex foot defects, a new anatomo-functional classification.

Santanelli di Pompeo F, Pugliese P, Sorotos M, Rubino C, Paolini G. Injury. 2015 Aug;46(8):1656-63. doi: 10.1016/j.injury.2015.05.002. Epub 2015 May 12.PMID: 26004168

Response to the above two comments:

The following discussion part has been added to the revised manuscript. It includes both requested references:

“Therefore, the ideal reconstruction of complex foot defects remains suboptimal. Santanelli di Pompeo et al. identified critical anatomical components that incorporate microvascular reconstruction to achieve favorable outcomes in demanding wounds [49]. They identified a structural classification including a bony platform, soft tissue envelope, and defect size as fundamental components when approaching demanding foot defects. Also, previous studies demonstrated a substantial improvement in the microcirculation of DFU via decompression of the plantar neurovascular bundle and tarsal tunnel release [50–52]. Furthermore, they addressed the importance of diabetic neuropathy as a cornerstone factor in DFU outcomes [52].” (page 13, lines 310–318).

New References:

49. Santanelli di Pompeo F, Pugliese P, Sorotos M, Rubino C, Paolini G. Microvascular reconstruction of complex foot defects, a new anatomo-functional classification. Injury. 2015;46: 1656–1663. doi:10.1016/j.injury.2015.05.002

50. Trignano E, Fallico N, Chen H-C, Faenza M, Bolognini A, Armenti A, et al. Evaluation of peripheral microcirculation improvement of foot after tarsal tunnel release in diabetic patients by transcutaneous oximetry. Microsurgery. 2016;36: 37–41. doi:10.1002/micr.22378

51. Dellon AL. The four medial ankle tunnels: a critical review of perceptions of tarsal tunnel syndrome and neuropathy. Neurosurg Clin N Am. 2008;19: 629–648, vii. doi:10.1016/j.nec.2008.07.003

52. Aszmann O, Tassler PL, Dellon AL. Changing the natural history of diabetic neuropathy: incidence of ulcer/amputation in the contralateral limb of patients with a unilateral nerve decompression procedure. Ann Plast Surg. 2004;53: 517–522. doi:10.1097/01.sap.0000143605.60384.4e

Reviewer 1

- the authors have done a fine job with the vascular component of this problem, but even though they mention diabetic neuropathy, they fail to discuss its role in their failure. This should be done

- Furthermore, it has been shown that tarsal tunnel decompression of the four medial ankle tunes can improve sensation, prevent ulcer and amputation. The authors should discuss the possibility of doing proximal nerve decompression at the same time as the amputaiton, and include these references. The decompression of the medial and lateral plantar vessels at the same time as the nerves may also improve blood flow.

Dellon, AL, The Four Medial Ankle Tunnels: A critical review of perceptions of tarsal tunnel syndrome and neuropathy, Neurosurg Clinics N America, 19:629-648, 2008.

Aszmann OC, Tassler PL, Dellon AL: Changing the natural history of diabetic neuropathy: Incidence of ulcer/amputation in the contralateral limb of patients with a unilateral nerve decompression procedure, Ann Plast Surg, 53:517-522, 2004.

Response: 

We thank the reviewer for this valid point. Peripheral neuropathy is often overlooked in our database due to the lack of objective tools for assessing neuropathy in vascular clinics and the misunderstanding of its inevitability in diabetic patients. In fact, 85.2% of the studied cohort are insulin-dependent patients with uncontrolled HbA1c in 69.1% of the subjects. This indirectly indicates the presence of long-standing diabetes and diabetic neuropathy in more than 2/3 of the studied cohort.

The above-requested references have been included in the manuscript, along with the discussion of peripheral neuropathy and nerve decompression as follows:

- First inclusion:

Also, previous studies demonstrated a substantial improvement in the microcirculation of DFU via decompression of the plantar neurovascular bundle and tarsal tunnel release [50–52]. Furthermore, they addressed the importance of diabetic neuropathy as a cornerstone factor in DFU outcomes [52]. (page 13, lines 315–318).

- Second inclusion: 

Furthermore, poor glycemic control is linked to an increased incidence of PAD and peripheral neuropathy. (page 14, line 326).

- Third inclusion:

Moreover, peripheral neuropathy is often overlooked in our database, and regrettably, the authors did not explore the impact of diabetic neuropathy on TMA healing and re-ulceration. (page 14, lines 328–330). 

- New References:

50. Trignano E, Fallico N, Chen H-C, Faenza M, Bolognini A, Armenti A, et al. Evaluation of peripheral microcirculation improvement of foot after tarsal tunnel release in diabetic patients by transcutaneous oximetry. Microsurgery. 2016;36: 37–41. doi:10.1002/micr.22378

51. Dellon AL. The four medial ankle tunnels: a critical review of perceptions of tarsal tunnel syndrome and neuropathy. Neurosurg Clin N Am. 2008;19: 629–648, vii. doi:10.1016/j.nec.2008.07.003

52. Aszmann O, Tassler PL, Dellon AL. Changing the natural history of diabetic neuropathy: incidence of ulcer/amputation in the contralateral limb of patients with a unilateral nerve decompression procedure. Ann Plast Surg. 2004;53: 517–522. doi:10.1097/01.sap.0000143605.60384.4e

Reviewer 2

- PAD is a significant etiology for development of ischemia hence TMA. There is a significant proportion of patients with low ABPI (mod/severe) in your cohort, but the revascularization rates are low. Would you be able to discuss more and comment? In addition, it will be interesting to see the effect of revascularization efforts amongst the sub-group of patients with low ABPI. Counter intuitively, your results showed that patients with TMA failure and major LEA had higher percentages of revascularization rather than those who healed. It’s likely these patients had more severe PAD, rather than the act of revascularization being a contributory factor.

Response: 

The authors thank the reviewer for this valuable comment. We have discussed the impact of PAD on TMA outcome and identified possible factors that might be linked to the relatively low revascularization rates among our cohort. Please refer to the sixth paragraph under the Discussion section of the manuscript on pages 12–13, lines 286–310. 

The following is the paragraph:

“There is skepticism in the existing literature on the benefit of revascularization in patients undergoing TMA. Some authors identified positive results with concomitant revascularization [22], while others denied recording any benefit [38]. Toursarkissian et al. [47] indicated that failed TMA is a multifactorial process, and failed revascularization is not a major event that predicts TMA failure. However, the authors included a selected group in which revascularization was performed in 79.5% of TMA patients with a mean toe pressure of 31 mmHg. Furthermore, the same authors indicated that the healing potential could not be predicted using angiographic findings. In our analysis, previous revascularization imposed a negative outcome on healing; revascularization was associated with poor healing at 1 year (P=0.031) and revision to proximal amputation (P=0.012) in patients that underwent TMA. This paradox is multifactorial and may be explained by the following facts regarding diabetic vasculopathy. First, it is likely that patients who underwent vascular procedures had a greater burden of arterial occlusive disease (ABI≤0.4), which would predispose to poor healing after TMA. In a study from China [38], the authors explored the outcomes of TMA in diabetic patients who were not candidates for revascularization and concluded that satisfactory results can be achieved in patients with limb ischemia undergoing TMA; however, only 11.1% of their study cohort had severe ischemia with ABI of ≤0.4. Second, the lack of a timely approach to revascularization when the mean time to revascularization was 34 days in the current data. Moreover, futile attempts to achieve inline flow in cases of severe occlusive disease of foot vessels and pedal arch (Desert foot) renders tibial interventions suboptimal to secure adequate tissue healing and is infrequently linked to physical deconditioning. Finally, vascular and endovascular interventions in patients with diabetes are technically demanding, with a significant early failure rate. As reported by Mueller et al. [36], vascular reconstruction before TMA is not always protective, as 57% of patients who required a more proximal amputation had prior vascular reconstructive surgery.”

-Also, please refer to our conclusion on page 16, lines: 391–393, as follows:

“Timely revascularization for patients with an ABI of ≤0.4 is of utmost interest when feasible, to allow effective positive outcome and avoid major future complications.”

- LRINEC score is often used for necrotising fasciitis. Would you be able to elaborate why it was adopted in your population?

Response: 

Again, we thank the reviewer for this important comment. We utilized the LRINEC score as a numerical method to predict outcomes in DFU and to extrapolate its utility as a prognostic tool from published data. The lack of validated tools incorporating substantial parameters that predict outcomes among DFU urged the authors to use the LRINEC score as a possible tool to aid informed consent and predict the outcome. Please refer to the 9th paragraph under the Discussion section on pages 14–15, lines 341–355, for a detailed explanation about adopting the LRINEC score in our population:

The paragraph is:

“In the current analysis, the LRINEC score was utilized as a numerical model to predict any trends in TMA outcomes as it combines multiple parameters related to wound healing. The counterintuitive inverse correlation between LRINEC score and healing potential can be easily explained by the fact that this score was principally developed to detect necrotizing fasciitis in soft tissue infections, and it lacks any parameter linked to vascular insufficiency, which is an important determinant of healing potential in DFU. In a recent study from Turkey [57], the authors explored the predictability of amputation or death in DFU according to the LRINEC score and showed that a score of ≥5 predicted amputations, while a score of ≥7 predicted mortality. In 2014, the WIFI system was developed by the Society for Vascular Surgery as a prognostic tool for predicting outcomes in patients with threatened diabetic limbs [58]. Interestingly, the WIFI system failed to predict the major LEA potential following TMA in a comparative study by Elsharif et al. [34]. Therefore, outcome predictive tools that incorporate parsimonious clinical, microbiological, and laboratory parameters, which are validated by data from vascular registries, should be developed to assist in making future decisions.”

- What are your neuropathy rates?

Response:

We thank the reviewer for this valid point. Peripheral neuropathy is often overlooked in our database due to the lack of objective tools for assessing neuropathy in vascular clinics and the misunderstanding of its inevitability in diabetic patients. This limitation has now been addressed in the revised manuscript as follows:

“Moreover, peripheral neuropathy is often overlooked in our database, and regrettably, the authors did not explore the impact of diabetic neuropathy on TMA healing and re-ulceration.” (page 14, lines 328–330).

- What is the time from TMA to full wound closure?

Response: 

In the studied cohort, 41/81 patients had complete healing of TMA. 9/41 healed within 3 months, 27/41 healed within 3-6 months, and 5/41 healed between 6–12 months.

- Any wound adjuncts utilised? (e.g., NPWT or any advanced wound care)

Response: 

Meticulous wound management was performed by our specialized tissue viability team, and the use of advanced wound healing products and negative pressure therapy was individualized according to case requirements. Six out of 81 patients received NPWT. The impact of NPWT was not discussed due to the limited number of patients that do not infer accurate data.

- What is your off-loading / footwear regime? Stump ulcer may be due to poor footwear, rather than an actual TMA failure

Response: 

Unfortunately, offloading footwear is not included in our insurance policy, and most patients cannot afford to buy these expensive yet essential assistive devices. Furthermore, the lack of awareness among patients and primary care providers remains an important obstacle to offloading footwear. Finally, UNICEF identified 5A’s that limit the use of assistive devices. These are Availability, Accessibility, Affordability, Adaptability, and Acceptability, especially in developing nations such as Jordan.

Lastly, Thanks for all your time and efforts in helping us to improve this manuscript. We hope that you now find this manuscript acceptable for publication in your esteemed journal.

---

## [Decision Letter · Decision Letter 1]

21 Oct 2022

Transmetatarsal Amputations in Patients with Diabetes Mellitus: A Contemporary Analysis from an Academic Tertiary Referral Centre in a Developing Community

PONE-D-21-26758R1

Dear Dr. ALJARRAH,

We’re pleased to inform you that your manuscript has been judged scientifically suitable for publication and will be formally accepted for publication once it meets all outstanding technical requirements.

Kind regards,

Tze-Woei Tan, M.D.

Academic Editor

PLOS ONE

Additional Editor Comments (optional):

Thank you for addressing the reviewers' comments.

Reviewers' comments:

Reviewer's Responses to Questions

**Comments to the Author**

1. If the authors have adequately addressed your comments raised in a previous round of review and you feel that this manuscript is now acceptable for publication, you may indicate that here to bypass the “Comments to the Author” section, enter your conflict of interest statement in the “Confidential to Editor” section, and submit your "Accept" recommendation.

Reviewer #1: All comments have been addressed

Reviewer #2: All comments have been addressed

2. Is the manuscript technically sound, and do the data support the conclusions?

Reviewer #1: Yes

Reviewer #2: Yes

3. Has the statistical analysis been performed appropriately and rigorously? 

Reviewer #1: Yes

Reviewer #2: Yes

4. Have the authors made all data underlying the findings in their manuscript fully available?

Reviewer #1: Yes

Reviewer #2: Yes

5. Is the manuscript presented in an intelligible fashion and written in standard English?

Reviewer #1: Yes

Reviewer #2: Yes

6. Review Comments to the Author

Reviewer #1: EXCELLENT JOB OF ANSWERING EACH REVIEWER'S COMMENTS

no additional revision is required

that is all i have to say

Reviewer #2: Congrats on your work. Thank you for your edits. I have no further comments.

7. PLOS authors have the option to publish the peer review history of their article (what does this mean?). If published, this will include your full peer review and any attached files.

Reviewer #1: **Yes: **A.Lee Dellon, MD, PhD

Reviewer #2: **Yes: **Zhiwen Joseph Lo

---

## [Editor Report · Acceptance letter]

25 Oct 2022

PONE-D-21-26758R1 

Transmetatarsal Amputations in Patients with Diabetes Mellitus: A Contemporary Analysis from an Academic Tertiary Referral Centre in a Developing Community 

Dear Dr. Aljarrah:

I'm pleased to inform you that your manuscript has been deemed suitable for publication in PLOS ONE. Congratulations! Your manuscript is now with our production department. 

Kind regards, 

on behalf of

Dr. Tze-Woei Tan 

Academic Editor

PLOS ONE